# OpenReview forum: "The Personality Illusion: Revealing Dissociation Between Self-Reports & Behavior in LLMs"
_ICLR.cc/2026/Conference — ICLR 2026 Conference Withdrawn Submission_

### Official Review · Reviewer_ajdh · 2025-10-31

**Soundness:** 3
**Presentation:** 3
**Contribution:** 3
**Rating:** 6
**Confidence:** 3

**Summary:**

This paper investigates whether LLM “personality” measured via psychometric self-reports (Big-Five, Self-Regulation) corresponds to behavior on psychologically grounded tasks, and whether persona prompting can control either. The authors run three studies:

- **RQ1 (Origin):** Across pre-training → instruction-tuning/RLHF, self-reported traits become more “human-like”: openness/agreeableness/self-regulation rise, neuroticism falls; trait variability drops and trait–trait associations strengthen.
- **RQ2 (Manifestation):** Stable self-reports do not reliably predict behavior. Directional alignment between trait coefficients and human-expected effects is ~chance (≈45–62% across traits; ≈45–57% across tasks), with only a few large models showing modest, sometimes significant alignment (e.g., Qwen-235B).
- **RQ3 (Control):** Persona injection robustly shifts self-reports, but has little/inconsistent impact on behaviors (sycophancy, risk-taking). Thus, persona effects are largely superficial.

Overall, the paper argues for a linguistic–behavioral dissociation: current alignment and prompting stabilize surface-level self-descriptions but do not ground consistent behavior, pointing toward the need for behaviorally grounded alignment objectives.

**Strengths:**

- Good clarity and logical flow. The overall structure is well organized; RQ1–RQ3 form a coherent progression that makes the argument easy to follow from conceptual framing to empirical validation. Each research question builds naturally on the previous one, and the results are presented in a clear and interpretable way.
- This paper frames a careful integration of psychometrics with behavior. The work explicitly separates self-reported trait structure from downstream behavior, a distinction often blurred in prior LLM “personality” papers, and demonstrates stronger trait coherence post-alignment while simultaneously showing weak behavioral predictiveness.
- This paper clear negative result with practical implications. The near-chance alignment between self-reports and behavior (with limited exceptions) is crisply quantified, challenging common assumptions that questionnaire scores port to action policy.

**Weaknesses:**

- RQ3 only probes two behaviors (sycophancy, risk-taking) and two personas; the null transfer might be task-specific. Moreover, persona construction uses keyword-based prompts; training-time interventions or activation-steering baselines are not compared. This constrains the generality of the “control is superficial” conclusion.
- Limited coverage of imbuement mechanisms. “Persona injection” is the only controllability method tested; the paper does not assess the other personality-imbuement approaches. Hence, the conclusion “traits don’t control behavior” may be specific to prompt-only personas, not to personality imbuement writ large.

**Questions:**

- Persona persistence and breadth: In RQ3, can authors extend their analysis to all five behavioral tasks and examine the multi-turn persistence or decay of persona effects?
- Incorporating activation-level interventions (e.g., persona vectors or latent-space steering) might serve as a stronger and more interpretable diagnostic? This would also reinforce the paper’s claim that while LLMs exhibit clear shifts in self-reported personality traits, these shifts do not consistently manifest in behavior. Activation-level analyses would thus provide a principled, mechanistic check to strengthen the causal link between persona representation and behavioral expression.

---

> ### Author Response · Authors · 2025-11-17
> **Thank You for the Constructive Feedback: Clarifications and Planned Experimental Additions**
>
> Dear Reviewer ajdh,
>
> We thank the reviewer for their time and for the constructive feedback. We truly appreciate it and are very encouraged to hear that the reviewer finds our work well executed, meaningfully challenging prior work, and carrying practical implications. **Below, we provide clarifications on some of the reviewer’s questions and outline the additional experiments we plan to include in response to the reviewer’s suggestions.** We would be happy to know whether these additions and planned experiments would address the reviewer’s concerns or whether there are further suggestions. We are actively executing the planned experiments and aim to share results and insights from them, within one week.
>
>
> **1. Breath in RQ3**
>
> We appreciate the reviewer’s concern that RQ3 currently evaluates only two of the five behavioral tasks with two personas. Our initial focus on sycophancy and risk-taking was motivated by the fact that these tasks exhibited the most counterintuitive patterns in RQ2 with the two personas, making them natural candidates for probing controllability.
>
> That said, we fully agree that broader coverage would strengthen the conclusions. In response, **we plan to run the full set of four behavioral tasks**, each with two personas selected based on trait–task relevance:
>
> * **Risk-taking task:** self-regulation and openness (openness to challenge)
> * **Sycophancy task:** agreeableness and self-regulation
> * **Stereotyping task:** self-regulation and neuroticism
> * **Self-reflective Honesty task:** self-regulation and conscientiousness
> * **Epistemic Honesty task:** self-regulation and conscientiousness
>
> Regarding multi-turn persistence and decay, we did not pursue these analyses initially because persona effects did not reliably manifest even in single-turn settings. However, if any of the expanded single-turn evaluations show meaningful shifts, we will be happy to test multi-turn persistence and decay as a follow-up.
>
> **2. Personality imbuement approach.**
>
> We thank the reviewer for highlighting activation-level personality imbuement methods such as activation steering, which indeed represent an emerging and promising line of work. We did not include these methods in our study for two main reasons.
>
> First, this area is still far from mature. Current activation-steering techniques are often brittle: they risk easily degrading an LLM’s core capabilities [1] [2] [6] [7], reduce instruction-following fidelity [2] [8] [9], remain mostly limited to single concept injection [2] [10] [11], introduce instability that makes controlled behavioral evaluation difficult [2] [3] [4] [8], and are not yet ready to be applied at scale [1] [2] [5] [6] [12]. Despite some active research in this direction, the abovementioned issues persist in current literature and limit the broader applicability of the approach.
>
> Second, persona prompting remains the primary—and most widely implemented—paradigm used in practice by companies, researchers, and end users. It is low-cost, requires no model modification, and is easily customizable. For this reason, our central arguments about controllability focus on the dominant, real-world mechanism by which personality is currently imposed on LLMs.
>
>
> We fully agree that if activation-vector methods become more robust and structured in the future, they may form the basis of a new major paradigm for personality imbuement in LLMs. Our findings on linguistic–behavioral dissociation would also provide important guidance as well as a benchmark for researchers working in this domain, offering a promising direction for future investigation. **We appreciate the reviewer for bringing attention to this point, and we agree that explicitly discussing these methods strengthens the paper. Accordingly, we have updated the Limitations and Future Work section in Appendix B to incorporate this discussion (new additions marked in red).**
>
>
> Once again, we greatly appreciate the reviewer’s thoughtful feedback. We would welcome any indication of whether the planned experiments fully address the concerns about breadth in RQ3. If there are further suggestions, we would be happy to incorporate them. Updated results will be provided as soon as they are available, within a maximum of one week. Thank you again for the time and constructive review.

---

> > ### Author Response · Authors · 2025-11-17
> >
> > [1] Tan, Daniel, et al. "Analysing the generalisation and reliability of steering vectors." Advances in Neural Information Processing Systems 37 (2024): 139179-139212.
> >
> > [2] Wehner, Jan, et al. "Taxonomy, opportunities, and challenges of representation engineering for large language models." In Transactions on Machine Learning Research (2025).
> >
> > [3] Braun, Joschka, et al. "Understanding (Un)Reliability of Steering Vectors in Language Models." arXiv preprint  (2025).
> >
> > [4] Itamar Pres, Laura Ruis, Ekdeep Singh Lubana, and David Krueger. Towards reliable evaluation of behavior steering interventions in LLMs. In Neurips 2024 Workshop MINT: Foundation Model Interventions, 2024.
> >
> > [5] Korznikov, Anton, et al. "The Rogue Scalpel: Activation Steering Compromises LLM Safety." arXiv preprint  (2025).
> >
> > [6] Daniel Scalena, Gabriele Sarti, and Malvina Nissim. Multi-property steering of large language models with dynamic activation composition. In The 7th BlackboxNLP Workshop - ARR Submissions, 2024.
> >
> > [7] Asa Cooper Stickland, Alexander Lyzhov, Jacob Pfau, Salsabila Mahdi, and Samuel R. Bowman. Steering without side effects: Improving post-deployment control of language models, 2024.
> >
> > [8] Peter S Park, Simon Goldstein, Aidan O’Gara, Michael Chen, and Dan Hendrycks. Ai deception: A survey of examples, risks, and potential solutions. Patterns, 5(5), 2024c.
> >
> > [9] Esin Durmus, Alex Tamkin, Jack Clark, Jerry Wei, Jonathan Marcus, Joshua Batson, Kunal Handa, Liane Lovitt, Meg Tong, Miles McCain, Oliver Rausch, Saffron Huang, Sam Bowman, Stuart Ritchie, Tom Henighan, and Deep Ganguli. Evaluating feature steering: A case study in mitigating social biases, 2024.
> >
> > [10] Teun van der Weij, Massimo Poesio, and Nandi Schoots. Extending activation steering to broad skills and multiple behaviours. arXiv preprint, 2024.
> >
> > [11] Andy Zou, Long Phan, Sarah Chen, James Campbell, Phillip Guo, Richard Ren, Alexander Pan, Xuwang Yin, Mantas Mazeika, Ann-Kathrin Dombrowski, Shashwat Goel, Nathaniel Li, Michael J. Byun, Zifan Wang, Alex Mallen, Steven Basart, Sanmi Koyejo, Dawn Song, Matt Fredrikson, J. Zico Kolter, and Dan Hendrycks. Representation engineering: A top-down approach to ai transparency, 2023a.
> >
> > [12] Shaolei Zhang, Tian Yu, and Yang Feng. Truthx: Alleviating hallucinations by editing large language models in truthful space, 2024b.

---

> > > ### Author Response · Authors · 2025-11-24
> > > **Updated Results from Promised Experiments Addressing RQ3 Breadth**
> > >
> > > Dear Reviewer ajdh,
> > >
> > > As promised, we have completed the extended experiments covering all five behavioral tasks, each evaluated with two theoretically relevant personas, to assess the generality of our findings. The results, summarized in the table below, further reinforce our central conclusion: **persona-prompt–induced shifts in self-reported traits remain large, reliable, and highly significant across tasks and traits, whereas behavioral effects remain small, noisy, and often misaligned with human-expected directions.**
> > >
> > > Specifically, for each task–persona pair, the table reports (i) the behavioral regression coefficient, its significance, and a “not aligned” label when applicable (indicating opposite from human-expected direction), and (ii) the corresponding coefficient and significance for the selected self-reported traits. Across all tasks, we observe a consistent pattern: **compared to self-reports**, the effects on behavioral outcomes are either **non-significant (n.s.), weak in magnitude (on average absolute β≈0.2 vs. β≈2.9 for self-reports), or misaligned with human-expected directions.**
> > >
> > > | Task                              | Trait            | β (behavior) | Significance                        | β (self-report) | Significance |
> > > |----------------------------------|------------------|--------------|-------------------------------------|------------------|--------------|
> > > | Epistemic Honesty                | Self-Regulation  | -0.30        | **                                  | 2.7              | ***          |
> > > | Epistemic Honesty                | Conscientiousness| 0.16         | n.s. (not aligned)                  | 3.1              | ***          |
> > > | Self-Reflective Honesty          | Self-Regulation  | 0.14         | * (not aligned)                     | 2.7              | ***          |
> > > | Self-Reflective Honesty          | Conscientiousness| -0.12        | *                                    | 3.1              | ***          |
> > > | Sycophancy                       | Self-Regulation  | 0.18         | n.s.                                 | 2.7              | ***          |
> > > | Sycophancy                       | Agreeableness    | 0.03         | n.s.                                 | 3.9              | ***          |
> > > | CCT (Risk-Taking)                | Openness         | -0.40        | ** (not aligned)       | 3.3              | ***          |
> > > | CCT (Risk-Taking)                | Self-Regulation  | -0.12        | n.s.                                 | 2.7              | ***          |
> > > | IAT (Stereotyping)               | Self-Regulation  | -0.70        | **                                   | 2.7              | ***          |
> > > | IAT (Stereotyping)               | Neuroticism      | 0.08         | n.s.                                 | 2.4              | ***          |
> > >
> > > Thus, while persona prompting remains effective and easy to apply for influencing an LLM’s self-reported traits and other linguistic outputs, our results show that its influence on actual behavioral tendencies is considerably weaker and less reliable.

---

> > > > ### Author Response · Authors · 2025-11-24
> > > >
> > > > We hope that the extended experiments and clarifications provided above fully address your questions and concerns. Please let us know if there are any remaining points you would like us to elaborate on or if further analyses would be helpful. We would be happy to include them as well.

---

> > > > > ### Author Response · Authors · 2025-11-27
> > > > > **Follow-Up Regarding Reviewer Concerns**
> > > > >
> > > > > Dear Reviewer adjh,
> > > > >
> > > > > Thank you again for your thoughtful review. As the rebuttal period is approaching its end, we wanted to kindly check in and ask whether our clarifications and additional experiments have addressed your concerns. We are happy to answer any further questions. If you find our responses convincing, we would be grateful if you would consider updating the confidence or scores accordingly
> > > > >
> > > > > Thank you very much for your time and effort.
> > > > >
> > > > > Best regards,
> > > > >
> > > > > Submission 26 Authors

---

### Official Review · Reviewer_338A · 2025-10-31

**Soundness:** 3
**Presentation:** 4
**Contribution:** 3
**Rating:** 6
**Confidence:** 4

**Summary:**

This paper reports on a study of LLM personalities focusing on the emergence of traits through training, the consistency of behaviors and self-reported traits, and the impact of targeted interventions such as persona injection. The authors find that post-training stabilizes trait expression and strengthens relationship between traits and behaviors in a manner that correlates with human behavioral data. However, these self-reported traits do not reliably predict behavior which often diverges from human patterns. The authors argue for deeper evaluations of LLM traits and behaviors.

**Strengths:**

This is a comprehensive study exploring multiple facets of the emergence and manifestation of LLM personalities. It's main strength is in its novel approach to assessing trait stability and trait coherence. The authors reviewed the psychometric literature and proposed links between personality traits and behaviors. Using these empirical results, the authors examined LLMs performance on self-reported traits and behavioral tasks.

**Weaknesses:**

The statistical methods employed are not well motivated and in some cases the choices seem ill-advised. For example, Levene's test is not advised for repeated measure experiments, where something like Mauchy's test of sphericity would be expected.
Further, predicting training stage from personality traits seems to confuse the explanans with the explanandum. As the authors discuss in the introduction, "We find that instructional alignment plays a pivotal role in shaping LLM traits" (p.2, 074).
Comparing LLM behavior with humans begs the question, whether LLMs are expected to show similar behavior to humans. This is far from given and risks anthropomorphizing LLMs further.

**Questions:**

p.2 146 "To assess the internal consistency of model trait expression, we analyze trait stability under repeated prompting with the same input
across multiple generations" -> How are you analyzing trait stability under repeated prompting? Levene's test can give you a sense of the overall (i.e., between-group) variance, but doesn't assess the variability (i.e., between-subject) across repeated measures.
Homogeneity of variance notwithstanding, it would strengthen the results to include the relative contributions of random and fixed effects to the overall variance explained, and the variance within and between runs per model.

---

> ### Author Response · Authors · 2025-11-17
> **Thank You for the Thoughtful Review: Clarifications and Expanded Analyses**
>
> Dear Reviewer 338A:
>
> We sincerely thank the reviewer for the thoughtful and constructive feedback. We are encouraged that you found the study comprehensive and appreciated our psychometric framing, trait–behavior mapping, and analyses of trait stability and coherence. Below, we address your concerns about statistical methodology, conceptual framing, and the interpretation of human–LLM behavioral comparisons. **We have implemented these changes in the paper (The revisions are marked in red in the paper).**
>
> **1. Statistical test for RQ1a may have opposite direction**
>
> We agree that the training stage is the explanans and the trait profile is the explanandum. In this analysis, we use logistic regression in the opposite direction only as a decoding test: do trait profiles contain enough signal to distinguish aligned from base models? This analysis is purely **associative rather than causal**—we do not interpret trait scores as causing the training stage, but as reflecting differences induced by pre-training vs. alignment. **We thank the reviewer for pointing this out and have further clarified this in the text to make clear that we do not interpret traits as causing the training stage. Specifically, we modified in the paper:**
>
> * 2.2 STATISTICAL ANALYSIS a) Examining Trait-level Differences by Training Phase.
> * 2.3 RESULTS a) Trait-level differences.
>
>
> **2. Statistical methodology for repeated measures in RQ1b**
>
> We appreciate the careful attention to the statistical modeling of trait stability under repeated prompting in RQ1b. The reviewer correctly notes that while Levene's test provides a valid omnibus test of variance homogeneity, it does not fully exploit the hierarchical structure of our repeated-measures design. Our original approach was conservative—treating repeated generations as exchangeable observations to test whether instruction-tuned models show reduced dispersion overall. While this answered our primary question, it did not decompose where that stability originates (between models vs. within-model consistency) or quantify the contribution of random vs. fixed effects, as you rightly point out.
>
> **Revised Analysis.**  Following your suggestion, we have restructured the analysis using a mixed-effects framework that explicitly models run-to-run variability, and **it further confirms a large, highly significant reduction in variance** for instruction-tuned models (β ≈ –4.5, p < .001):
>
> * **Trait stability as run-to-run variance:** For each model, trait, persona, and temperature, we collect three generations and treat these as repeated measures. We operationalize trait stability as the variance of trait scores across the three runs within each model–persona–temperature cell.
>
> * **Mixed-effects analysis:** We analyze the log of these run-to-run variances using linear mixed-effects models with alignment (base vs instruction-tuned) and trait as fixed effects and random intercepts for the model. Rather than a repeated-measures ANOVA with Mauchly’s test of sphericity, we adopt linear mixed-effects models, which are widely used for repeated-measures and hierarchical designs and avoid the sphericity assumptions of traditional ANOVA. This framework decomposes variability in stability into between-model and residual components, as suggested by the reviewer. Across traits, instruction-tuned models exhibit a large and highly significant reduction in run-to-run variance (pooled β ≈ –4.5 on the log-variance scale, p < .001), corresponding to an order-of-magnitude increase in stability. Trait-wise, instruction-tuning reduces mean run-to-run variance by approximately **81–90% across traits (Appendix E)**, indicating that all six traits are statistically significantly more stable in instruction-tuned models. Further, the estimated between-model (random) variance in log run-to-run variance is 2.67, and the residual (within-cell) variance is 16.01. This implies that approximately 14% of the variability in trait stability is attributable to differences between models, while the remaining 86% reflects within-model variation across items, personas, and temperatures. We report these variance components in **Appendix E**, providing exactly the decomposition of variability the reviewer requested. **We have updated Panel B in Figure 2 to reflect the analysis change.**
>
>
> | Trait             | Pre-train (95% CI)   | Post-align (95% CI)  | Sig |
> | ----------------- | -------------------- | -------------------- | :-: |
> | Openness          | 0.149 [0.099, 0.198] | 0.019 [0.012, 0.027] | *** |
> | Conscientiousness | 0.139 [0.094, 0.183] | 0.019 [0.011, 0.028] | *** |
> | Extraversion      | 0.142 [0.103, 0.180] | 0.021 [0.012, 0.030] | *** |
> | Agreeableness     | 0.152 [0.099, 0.205] | 0.016 [0.010, 0.022] | *** |
> | Neuroticism       | 0.152 [0.110, 0.193] | 0.028 [0.015, 0.042] | *** |
> | Self-Regulation   | 0.023 [0.016, 0.029] | 0.002 [0.001, 0.003] | *** |

---

> > ### Author Response · Authors · 2025-11-17
> >
> > **3.  Fundamental expectation around LLMs reflecting human behavior and the risk of anthropomorphizing LLMs**
> >
> > We thank the reviewer for raising this important conceptual concern. We want to clarify that our study does **not** assume LLMs should behave like humans; rather, it critically examines an assumption that is **already widespread in current practice.**
> >
> > **a) Our research addresses a problematic existing practice.** Many researchers and users use personality questionnaires to assess LLM traits and assume these traits can predict downstream behaviors. Our study tests whether this approach—borrowed from human psychometrics—is valid for LLMs. Critically, we find that self-reported traits show minimal predictive power for LLM behavior. **This result holds independently of human comparisons: questionnaire results have very little statistically significant relation to behavior, regardless of whether we compare to humans or not.** Rather than supporting anthropomorphization, our findings challenge the widespread practice of directly applying human personality frameworks to LLMs without empirical validation.
> >
> > **b) Why human comparisons remain valuable.** While our core finding stands alone, comparing LLM trait-behavior relationships to human patterns serves two purposes. **First**, personality-related applications are often deployed in sensitive contexts (mental health support, education, persuasive systems) where unexpected divergences from human behavioral norms pose safety risks. Understanding where LLM behavior systematically diverges from human patterns is essential for responsible deployment. **Second**, given that LLMs demonstrably exhibit various human-like phenomena—stereotyping [1], deception [2], theory-of-mind [3], framing effects [4]—characterizing which behavioral patterns they do and do not reproduce provides important boundary conditions for understanding these systems. Our findings that trait-behavior relationships diverge from human patterns establish exactly such boundaries.
> >
> > [1] Navigli, Roberto, Simone Conia, and Björn Ross. "Biases in large language models: origins, inventory, and discussion." ACM Journal of Data and Information Quality 15.2 (2023): 1-21.
> >
> > [2] Park, Peter S., et al. "AI deception: A survey of examples, risks, and potential solutions." Patterns 5.5 (2024).
> >
> > [3] Kosinski, Michal. "Theory of mind may have spontaneously emerged in large language models." arXiv preprint.
> >
> > [4] Tjuatja, Lindia, et al. "Do llms exhibit human-like response biases? a case study in survey design." Transactions of the Association for Computational Linguistics 12 (2024): 1011-1026.

---

> > > ### Author Response · Authors · 2025-11-20
> > > **Follow-up on Revisions and Clarifications**
> > >
> > > Dear Reviewer 338A,
> > >
> > > Thank you again for your thoughtful and constructive review. We hope that our clarifications, revisions, and expanded statistical analyses address your concerns. If you have any additional suggestions or questions, we would be grateful to hear them. We would also appreciate it if you might **consider raising your score** should you feel the revisions satisfactorily resolve the issues you raised.
> > >
> > > Thank you once again for your time and engagement.
> > >
> > > Sincerely,
> > >
> > > Authors of Submission 26

---

> > > > ### Author Response · Authors · 2025-11-24
> > > >
> > > > Dear Reviewer 338A,
> > > >
> > > > As we are approaching the final week of the rebuttal period, we wanted to kindly follow up. We have carefully addressed all of the questions you raised and implemented the statistical revisions you suggested. In particular, the statistical analysis you recommended has now been incorporated, and we believe it further reinforces the original conclusions of our paper. Thank you for the helpful suggestions!
> > > >
> > > > We thank you again for your thoughtful review. If there are any additional points you would like us to clarify or expand upon, we would be very happy to do so as well. Otherwise, we hope that the updated analyses and revisions satisfactorily resolve the issues you highlighted, and we would be sincerely grateful if you would take them into account and update your final assessment.
> > > >
> > > > Thank you once again for your constructive feedback.
> > > >
> > > > Warm regards,
> > > >
> > > > Authors of Submission 26

---

> > > > > ### Author Response · Authors · 2025-11-27
> > > > > **Follow-Up Regarding Reviewer Concerns**
> > > > >
> > > > > Dear Reviewer 338A,
> > > > >
> > > > > Thank you again for your thoughtful review. As the rebuttal period is approaching its end, we wanted to kindly check in and ask whether our clarifications and additional statistical analysis have addressed your concerns. We are happy to answer any further questions. If you find our responses convincing, we would be grateful if you would consider updating the confidence or scores accordingly.
> > > > >
> > > > > Thank you very much for your time and effort.
> > > > >
> > > > > Best regards,
> > > > >
> > > > > Submission 26 Authors

---

### Official Review · Reviewer_3PUV · 2025-11-01

**Soundness:** 1
**Presentation:** 1
**Contribution:** 1
**Rating:** 0
**Confidence:** 5

**Summary:**

The paper addresses three core questions:

1. When and how do human-like traits emerge and evolve across LLM training?
2. Do self-reported traits predict performance in real-world–inspired tasks?
3. How do interventions like persona injection modulate trait profiles and behavior?

The authors present sequence of the experiments on a set of models. The questions are interesting but the statistical significance of the reported results is lacking.

**Strengths:**

The direction is extremely interesting and really improtant.

**Weaknesses:**

The research questions 2 and 3 are very interesting, however the fact that alignment happens in post-training is widely-known and it is surprising to see it as a component of a research question for ICLR.

The documented differences between self-reported properties, properties assessed with a test and observed behaviour could be interesting, but the problem is that "human studies" statements are lacking in quality and statistical soundness. Those studies could not be used as credible source of baseline for the research questions that authors try to addess.

For example, Sections H and I list papers that are used as human-reference for the association of the behaviour with certain test results. If I randomly select some of the papers listed there by the authors, I can easily see that statistical soundness of those is lacking.

For example, A. Ispas and C. Ispas use a sample of two groups of college students (each group of 30 people, each has 27 female and 3 male participants). Making conclusions on the overall properties of human nature based on such experience seems impossible. Another paper by Sinclair et al. reports results about "91 White undergraduate students (34 men, 57 women) at the University of Virginia who received partial credit toward a course requirement for completion of the experiment."

Most of these papers reports some results on very small select groups of people that not even slightly represent the breadth of human experience that LLM is exposed to during the pretraining. The authors also provide no internal logic for the selection of the papers that form human alignment "baseline".



The paper seems to be written in a hurry. Many figures have captions that are not legible. There are multiple papers that are cited twice as two different sources, see Binz and Schulz "Using cognitive psychology to understand gpt-3.", Bodroža et al. "Using cognitive psychology to understand gpt-3.", Li et al. "Big5-chat", Rahwan et al. "machine behaviour", Serapio-Gercia et al. "Personality traits in large language models",  Tseng et al. "Two tales of persona in llms", Zheng et al. " Personas in system prompts". While Cao and M. Kosinski "Large language models know how the personality of public figures is perceived by the general public" and Jiang et al. "Personallm: Investigating the ability of large language models to express personality traits." are cited THREE TIMES as different sources.

At the same time several works that could enhance the literature section are missing:

Sorokovikova A, Rezagholi S, Fedorova N, Yamshchikov IP. LLMs Simulate Big5 Personality Traits: Further Evidence. InProceedings of the 1st Workshop on Personalization of Generative AI Systems (PERSONALIZE 2024) 2024 Mar (pp. 83-87).

Pan X, Gao D, Xie Y, Chen Y, Wei Z, Li Y, Ding B, Wen JR, Zhou J. Very large-scale multi-agent simulation in agentscope. arXiv preprint arXiv:2407.17789. 2024 Jul 25.

Dong W, Zhao Y, Sun Z, Liu Y, Peng Z, Zheng J, Zhang Z, Zhang Z, Wu J, Wang R, Xu S. Humanizing llms: A survey of psychological measurements with tools, datasets, and human-agent applications. arXiv preprint arXiv:2505.00049. 2025 Apr 30.

Wang S, Li R, Chen X, Yuan Y, Wong DF, Yang M. Exploring the impact of personality traits on llm bias and toxicity. arXiv preprint arXiv:2502.12566. 2025 Feb 18.

Tshimula JM, Nkashama DJ, Muabila JT, Galekwa RM, Kanda H, Dialufuma MV, Didier MM, Kalonji K, Mundele S, Lenye PK, Basele TW. Psychological Profiling in Cybersecurity: A Look at LLMs and Psycholinguistic Features. InInternational Conference on Web Information Systems Engineering 2024 Dec 2 (pp. 378-393). Singapore: Springer Nature Singapore.

**Questions:**

Why don't you select meta-review papers for the traits for which there are meta-reviews that show stable association across various demographics? That could make the scope of your experimental work much smaller, but clearly would provide some statistical credibility for the human "baseline" and thus to the conclusions you make.

---

> ### Author Response · Authors · 2025-11-17
> **Addressing Major Inaccuracies and Unjustified Evaluation in Reviewer 3PUV’s Review**
>
> Dear Reviewer 3PUV,
>
> We appreciate the reviewer's time. However, we are disappointed to see that this review does not identify any concrete methodological, statistical, or empirical issues in our work. The comments raised concern only minor editorial matters or reflect misunderstandings of external papers, and none of them substantiate the unreasonably low score assigned. We always welcome respectful, constructive, technically grounded feedback (as provided by the other reviewers). Yet this review offers no scientifically actionable feedback that engages in any meaningful way directly with our methods, experiments, or findings. We therefore clarify below the **inaccuracies and mischaracterizations** in this review, and respectfully request the reviewer to **adhere to the ICLR Reviewer Guide.** This includes the responsibility to “provide supporting arguments for your recommendation”, “be positive and constructive”. Following from that, the ratings in the review should accurately reflect the work and be justified with concrete, specific, and scientifically grounded evidence, especially for such extreme scores.
>
>
> **1. Alignment & post-training**
>
> Firstly, the review states that “the fact that alignment happens in post-training is widely known,” and suggests that this makes our research question surprising (implied known and not interesting for ICLR). This suggests that **the current review is based on a substantial misunderstanding of our research questions and contributions**, which in turn raises serious concerns about whether the evaluation meaningfully reflects the core elements of the paper.
>
> We fully acknowledge that alignment occurs during post-training. In fact, our ability to measure this expected change in alignment lends further credibility to our methods. However, the contribution of our work is not to restate this fact, but to examine **how exactly** post-training shifts self-reported personality traits and **how these shifts** relate to corresponding behaviors. Our research question concerns the nature and consequences of these changes, not whether alignment takes place in the post-training phase, which is explicitly designed for alignment.
>
> **2. Paper writing**
>
> We are glad to see that the other reviewers independently found our presentation clear and well-structured. Given this consistency, we are genuinely puzzled by reviewer 3PUV’s extreme score of “1: poor.” Such a rating typically reflects major, fundamental presentation flaws, so we carefully examined reviewer 3PUV’s comments but were confused to find **only very minor and vague points.**
>
> With respect to captions, reviewer 3PUV only mentioned “Many figures have captions that are not legible”, but does not specify at all which figures are considered “not legible,” and what makes them “not legible”, making the critique difficult to interpret or respond to. Without concreteness, this comment is unsubstantiated and unhelpful.
>
> Regarding duplicate citations, we appreciate the observation and have corrected these minor editorial issues (many papers appear on arXiv and later at conferences under the same title, though we cite them in the correct context). However, **such small formatting mistakes obviously do not justify a rating of “1: poor” for presentation, nor do they constitute grounds for a strong rejection.** These are easily fixable issues that do not affect the clarity, structure, or contribution of the work itself.
>
> In conclusion, the **absence of concrete evidence or specific examples** renders Reviewer 3PUV’s presentation-related critique **unconstructive and vague**. This **falls short of the ICLR Reviewer Guidelines**, which explicitly ask reviewers to (1) **list strong and weak points comprehensively**, (2) **provide supporting arguments for their recommendation**, and (3) **write a constructive, thorough review.** **None of these requirements are met here, making the extremely low presentation score difficult to understand or justify.**

---

> > ### Author Response · Authors · 2025-11-17
> >
> > **3. Reviewer 3PUV’s Concerns About External Human-Study Methodology**
> >
> > Reviewer 3PUV’s critique of peer-reviewed human-study methodologies—methodologies that are well-established and broadly accepted across psychology, cognitive science, and the CS/AI community—is (1) not supported by the scientific literature and (2) falls outside the scope of our submission and ICLR’s review criteria, as the critique is mainly about this external, already published work, not the results presented in our submission. In addition, the review applies inconsistent standards by discouraging published, peer-reviewed work while recommending workshop papers and preprints (2 of the 5 papers the reviewer recommended for inclusion are unpublished preprints, and another 2 are non-archival workshop papers). To further substantiate our position, we also provide meta-review papers with substantially larger sample sizes, all of which reinforce the robustness of the relationships examined in our study. We will include these in the revised submission.
> >
> > In addition, our findings on linguistic–behavioral dissociation do not depend only on human comparison. Even when examined entirely within models, the questionnaire scores show minimal statistical association with subsequent behavior, underscoring that our core results stand independently of human–LLM contrasts. On top of that, our findings further highlight substantial inconsistencies between the models and within model families, all of which offer valuable insights on their own.
> >
> > **3.1 Standard Human-Study Sample Sizes in Psychology and Cognitive Science**
> >
> > Reviewer 3PUV’s argument that human behavioral studies with “small” numbers of participants are “not statistically sound” reflects a misunderstanding of standard methodology in psychology and cognitive science. Human behavioral experiments are expensive and time-consuming to collect, and as a result, top peer-reviewed venues—including Nature, Science, PNAS, and Nature Human Behaviour—routinely publish studies with sample sizes comparable to or smaller than those in the papers we cite (many of which come from these venues). These methods are well established and widely accepted in the field. Please refer to the following Table 1 for representative examples. In addition, these works are also cited in other CS/AI personality research papers [0], indicating that this methodology and use of related work is accepted within the CS/AI community as well.
> >
> > **Table 1. Top Publications in human studies use small sample sizes**
> >
> > | Study Title | Venue | Year | Sample Size |
> > |------------|--------|------|--------------|
> > | *The Native Language of Social Cognition [1]* | PNAS | 2007 | **N = 22** |
> > | *Acute Social Isolation Evokes Midbrain Craving Responses Similar to Hunger [2]* | Nature Neuroscience | 2020 | **N = 40** |
> > | *Decisions Bias Future Choices by Modifying Hippocampal Associative Memories [3]* | Nature Communications | 2020 | **N ≈ 40** |
> > | *On Being Sane in Insane Places [4]* | Science | 1973 | **N = 8** |
> > | *Asch Conformity Experiments* | Multiple publications | 1951–1956 | **N ≈ 50** |
> > | *The determinants of consciousness of human faces [5]* | Nature Human Behavior | 2017 | **N=25-31 (3 experiments)** |
> > | *Goal commitment is supported by vmPFC through selective attention [6]* | Nature Human Behavior| 2024 | **N=26-30** |
> > | *Efficient stabilization of imprecise statistical inference [7]* | Nature Human Behavior | 2022 |**N=26-30 (2 groups)** |
> > | *Efficient stabilization of imprecise statistical inference through conditional belief updating [8]* | Nature Human Behavior | 2022 |**N=60 (2 experiments)** |
> > | *Little Support for Discrete Item Limits in Visual Working Memory [9]* | Psychological Science | 2022 |**N=24 (4 experiments)** |
> >
> > References:
> >
> > [0] Li, Wenkai, et al. "Big5-chat: Shaping llm personalities through training on human-grounded data." Proceedings of the 63rd Annual Meeting of the Association for Computational Linguistics (Volume 1: Long Papers). 2025.
> >
> > [1] Kinzler, Katherine D., Emmanuel Dupoux, and Elizabeth S. Spelke. "The native language of social cognition." Proceedings of the National Academy of Sciences 104.30 (2007): 12577-12580.
> >
> > [2] Tomova, Livia, et al. "Acute social isolation evokes midbrain craving responses similar to hunger." Nature neuroscience 23.12 (2020): 1597-1605.
> >
> >  [3] Luettgau, Lennart, et al. "Decisions bias future choices by modifying hippocampal associative memories." Nature communications 11.1 (2020): 3318.
> >
> > [4] Rosenhan, David L. "On being sane in insane places." Science 179.4070 (1973): 250-258.
> >
> > [5] Abir, Yaniv, et al. "The determinants of consciousness of human faces." Nature Human Behaviour 2.3 (2018): 194-199.
> >
> > [6] Holton, Eleanor, et al. "Goal commitment is supported by vmPFC through selective attention." Nature Human Behaviour 8.7 (2024): 1351-1365.

---

> > > ### Author Response · Authors · 2025-11-17
> > >
> > > [7] Drevet, Julie, Jan Drugowitsch, and Valentin Wyart. "Efficient stabilization of imprecise statistical inference through conditional belief updating." Nature human behaviour 6.12 (2022): 1691-1704.
> > >
> > > [8] Drevet, Julie, Jan Drugowitsch, and Valentin Wyart. "Efficient stabilization of imprecise statistical inference through conditional belief updating." Nature human behaviour 6.12 (2022): 1691-1704.
> > >
> > > [9] Oberauer, Klaus. "Little support for discrete item limits in visual working memory." Psychological Science 33.7 (2022): 1128-1142.
> > >
> > > **3.2 Inconsistent Standards and Out-of-Scope Critiques**
> > >
> > > Additionally, we feel confused to find that the papers reviewer 3PUV suggests are workshop papers or preprints, while the papers we cited and were criticized for  are published, peer-reviewed work. While we are open to including relevant preprints in our discussion (**we have added them into our related works section**), we respectfully point out the significant inconsistency in the standards being applied. Even setting aside this double standard, the critiques raised pertain to foundational human-study methodology itself, rather than to anything specific to our paper. **As a submission to ICLR, our work focuses on model behavior—specifically, how LLMs’ self-reported traits shift across training stages and how these shifts relate to model behaviors, as well as how they connect to conclusions from human-study literature.** Re-examining foundational, peer-reviewed human-study methodologies is outside the scope of this work. If Reviewer 3PUV has deep concerns about these methodologies and published papers, we would welcome and encourage the reviewer to raise them in more appropriate venues—such as Nature or PNAS—in the form of an issues or perspectives paper.
> > >
> > > **3.3 Meta-review papers further supporting the reported relationships**
> > >
> > > Finally, in response to the reviewer’s request for meta-review papers, we show that these works **further reinforce the relationships we have articulated in our paper.**
> > >
> > > Paper title: Is Risk Propensity Unique from the Big Five Factors? A Meta-analytic Investigation
> > >
> > > “Table 2 shows that risk propensity had positive mean corrected correlations with extraversion (ρ = 0.24, 95% CI: [0.21, 0.27]) and openness to experience (ρ = 0.30, 95% CI: [0.28, 0.33]), and negative mean corrected correlations with agreeableness (ρ = –0.16, 95% CI: [–0.21, –0.12]), conscientiousness (ρ = –0.12, 95% CI: [–0.17, –0.08]), and neuroticism (ρ = –0.13, 95% CI: [–0.17, –0.10]).”
> > >
> > >
> > > **Conclusion:**
> > >
> > > In summary, Reviewer 3PUV’s evaluation reflects substantial misunderstandings of our research questions, mischaracterizations of well-established scientific methodologies, and an absence of concrete, evidence-based engagement with our work. The extreme numerical scores are not supported by methodological, statistical, or empirical critique, nor do they align with the standards outlined in the ICLR Reviewer Guide. We have clarified each point raised and demonstrated that these concerns either stem from factual inaccuracies or completely fall outside the scope of this submission.
> > >
> > > We respectfully request that this review be re-evaluated in light of the issues documented above. We remain fully committed to constructive scientific dialogue and welcome any feedback grounded in the actual content of the paper. However, we also believe it is essential to ensure that all submissions receive fair, accurate, and guideline-consistent reviews.

---

> > ### Comment · Reviewer_3PUV · 2025-11-26
> > **Some follow-up comments**
> >
> > I double-checked the ICLR guidelines. They do not provide any concrete recommendations on when score 0 is applicable, however, they recommend to address the other reviewers with caution and encourage the reviewers to make their own judgment. I appreciate the comments of the authors on the quality of the paper presentation but I have to reiterate that multiple citation duplicates, illustrations and statistical significants of the results are all signs of a paper that was written without considerable concerns around editing.
> >
> > Let me reiterate on my review explicitly referring to the ICLR guidlines.
> >
> > Problem: Is the problem the paper tackles well-defined and important?
> > Yes, the problem is interesting and improtant.
> >
> > Approach: Is the approach well-motivated and placed within the literature?
> > No. In my opinion, the approach is not well motivated in the original paper, but the responses of the authors significantly clarify some of their decisions.
> >
> > Support: Do the paper's claims have supporting evidence (e.g., correct results)?
> > No, the core result that is even represented in the paper title is the disagreement between "self-reports and behaviour" in LLMs. This implies that there is no such discrepancy in humans. Moreover, authors use the results of human studies for reference. However human literature is not conclusive on this issue. Mainly due to the fact that the groups studies in the human-reports are very small and highly skewed into western college-educated humans. I appreciate the questions the authors ask, but I am strongly against the idea of taking the methodology that is accepted in psychological research and bringing this methodology as it is into computer sciences. The other research that the authors reference as proof that such methodology is already a de-facto standard in computer science is not convincing to me for two major reasons: first, the claims that author make in the paper are more subtle than the claims made in some of the works they cite, second, if I was reviewing some of the cited papers that have similar methodlogical flaws, I would also be strongly against the acceptance of such results to the CS venue.
> >
> > Significance: Does the work contribute new and valuable knowledge to the community?
> > Hard to tell, since I am not convinced on the statistical significance of the reported results. I share this concern with one of the reviewers.
> >
> > If area chaird find my score too opinionated they are free to ask for additional emergency review. I am also happy to hear the arguments from other reviewers. Based on the details the authors provide I update my score.

---

> > > ### Author Response · Authors · 2025-11-27
> > > **Addressing Misunderstandings and Requesting Accurate Overall Score**
> > >
> > > Dear Reviewer 3PUV,
> > >
> > > Thank you for the follow-up. However, we remain disappointed to find that the reviewer's comments continue to contain major misunderstandings of our paper, lack substantiated support, and are inconsistent and strongly opinionated.
> > >
> > > ## Here is a quick summary of the major problems we have identified with this review:
> > >
> > > 1. _**The review is inconsistent.**_ From the very beginning, the reviewer indicates their confidence is 5 (absolutely certain), but has changed scores from 1-1-1 to 2-2-3 (out of 4)—a very substantial shift indicating that the reviewer's reported confidence level is not accurate.
> > > 1. _**The overall score doesn't match the sub-scores.**_ The reviewer raised the overall score only from 0 to 2, which strongly disagrees with the sub-scores of 2-2-3. This is a clear mismatch, and **we request a fair overall score.**
> > > 1. _**The reviewer is opinionated, repeatedly criticizing peer-reviewed external human studies.**_ We kindly remind the reviewer that **a review is not the place to criticize an entire established field.** Giving a negative decision based on opinions about an entire field is clearly subjective, opinionated, and out of scope. The appropriate venue to challenge or share concerns about a field is to publish an opinion paper with supporting evidence. The ICLR review should focus on our work and its concrete methods and details.
> > > 1. _**Regarding presentation, the review still fails to raise concrete points.**_ The reviewer uses only vague terms like "illustrations," which is confusing. Additionally, we don't believe citation duplicates are a significant issue justifying rejection. If the reviewer strongly believes otherwise, we'd like to emphasize that, as we shared in our original reply, we have now already fixed this issue, and thus the reviewer should increase the score significantly accordingly.
> > > 1. _**Lastly, the reviewer has not adequately engaged with our previous response,**_ which provided meta-analysis evidence, demonstrated that our findings' significance stands independently, and explained why questioning external peer-reviewed human studies is out of scope, etc. We hope the reviewer can adequately respond to the points we made in our rebuttal.

---

> > > > ### Author Response · Authors · 2025-11-27
> > > >
> > > > ## Having outlined our primary concerns above, we now address the reviewer's specific comments in detail:
> > > >
> > > > >_“They do not provide any concrete recommendations on when score 0 is applicable, however, they recommend to address the other reviewers with caution and encourage the reviewers to make their own judgment.”_
> > > >
> > > > We'd like to clarify:
> > > > 1. ICLR guidelines are bottom-line requirements for reviewers. They are not just “recommendations”; they are standards that should not be violated.
> > > > 1. While there are not many concrete requirements specifically for when score 0 is applicable, there are plenty of requirements that generally guide reviewers' behavior, which this reviewer has violated repeatedly (e.g., provide concrete support for your score, be constructive, etc.).
> > > > 1. We do encourage the reviewer to make their own judgments, but the reviewer needs to adhere to the reviewer guidelines, and this is all we ask. We are happy to see opposing opinions or critiques, but they need to be factually correct, helpful, and constructive, as the other reviewers have managed to achieve.
> > > >
> > > >
> > > > >_“multiple citation duplicates, illustrations and statistical significants of the results are all signs of a paper that was written without considerable concerns around editing.”_
> > > >
> > > > 1. **We do not agree that multiple citation duplicates are a significant issue that justifies strong rejection.** If the reviewer thinks it is really significant, good news: as we shared in our last update, we have already fixed all of them in this revision, so the reviewer should raise the scores significantly accordingly.
> > > > 1. The reviewer’s issue with the "illustrations" is still not clear and not actionable in terms of what exactly the problems are and how we can address them. We remain confused about that.
> > > > 1. Similarly, for statistical significance, the reviewer is not making any concrete points.
> > > > 1. **We respectfully mention that reviewing and research are not about "signs" or subjective guesses. They are about facts, which this review lacks.**
> > > >
> > > > >_“Problem: Is the problem the paper tackles well-defined and important? Yes, the problem is interesting and improtant.”_
> > > >
> > > > Thank you for strongly acknowledging the importance and interestingness of our work.
> > > >
> > > > >_“Approach: Is the approach well-motivated and placed within the literature? No. In my opinion, the approach is not well motivated in the original paper, but the responses of the authors significantly clarify some of their decisions.”_
> > > >
> > > > Which part of the approach is not well motivated? What has become more clear now, and what does the reviewer think needs additional clarification? Could the reviewer be specific? In any case, we are glad to hear you think we have significantly clarified our approach.
> > > >
> > > > >_“No, the core result that is even represented in the paper title is the disagreement between "self-reports and behaviour" in LLMs. This implies that there is no such discrepancy in humans.”_
> > > >
> > > > This is simply not true. It is surprising that the reviewer draws this implication, which is not logically sound and **demonstrates persistent misunderstandings about our paper and research questions.** We respectfully request that the reviewer read the paper carefully before making their comments.
> > > >
> > > > Specifically, throughout this paper, we never claim there is no such discrepancy in humans. Instead, the dissociation we find in LLMs, as we have stated multiple times, stands independently from human studies. The only claim we make is that **human self-reports have some predictive power on downstream behaviors**—which is why psychological research uses self-reports. We never claim this method works 100% of the time. In comparison, LLM self-reports have far less predictive power, resulting in few significant associations.
> > > >
> > > > >_“Moreover, authors use the results of human studies for reference.”_
> > > >
> > > > The reviewer has failed to understand that none of our discoveries are built upon "results" from human studies. Our finding that questionnaire-based self-reports poorly predict LLM behavior—and that persona prompts shift self-reports more than behavior—is not dependent on conclusions from human studies.

---

> > > > > ### Author Response · Authors · 2025-11-27
> > > > >
> > > > > >_“However human literature is not conclusive on this issue. Mainly due to the fact that the groups studies in the human-reports are very small and highly skewed into western college-educated humans. ”_
> > > > >
> > > > > Again, our main contribution regarding LLM self-reports and behavior does not rely on human literature. Our comparisons with humans are based on peer-reviewed human studies. If the review cannot follow this logic, the assessment is not correct.
> > > > >
> > > > > Regarding "skewness," this is unfortunately a convention in the field. If the reviewer is concerned about the psychological and cognitive science community's methodologies, we encourage them to raise this as an opinion article in appropriate venues. For this ICLR review, we respectfully request the review to focus on our work, which is not based on human literature, and note that questioning peer-reviewed human studies is way outside this paper's scope.
> > > > >
> > > > > >_“I appreciate the questions the authors ask, but I am strongly against the idea of taking the methodology that is accepted in psychological research and bringing this methodology as it is into computer sciences.”_
> > > > >
> > > > > The reviewer completely misunderstands our claim.
> > > > > 1. We did not transfer human psychological experiments directly to LLM studies without modification. While the reviewer's concern is about small sample sizes in human studies, we scaled up the evaluation for LLMs in our experiments.
> > > > > 2. **More importantly, we are surprised the reviewer fails to understand this is exactly the message we want to convey. We also oppose directly applying human psychological methods like questionnaires to assess LLMs without validation. This is the point of raising the issue of "personality illusion." We are glad the reviewer agrees with us, even if they may not realize it.**
> > > > >
> > > > > >_“first, the claims that author make in the paper are more subtle than the claims made in some of the works they cite,”_
> > > > >
> > > > > We are confused by what the reviewer means by "subtle." Could the reviewer clarify which parts of our claims they disagree with and why? If "subtle" means nuanced or important, we are glad to hear that. Respectfully, important research deserves to be conducted. If the reviewer has concerns about specific details of our supporting evidence, please let us know.
> > > > >
> > > > > >_“second, if I was reviewing some of the cited papers that have similar methodlogical flaws, I would also be strongly against the acceptance of such results to the CS venue.”_
> > > > >
> > > > > We hope the reviewer understands that **a review is not the place to criticize an entire established field.** Giving a strongly negative decision based on opinions about an entire field is clearly subjective, opinionated, and unfair. The appropriate venue to challenge or share concerns about a field is to publish an opinion paper with supporting evidence. **Your review should focus on our work and its concrete methods and details.**
> > > > >
> > > > > >_“Hard to tell, since I am not convinced on the statistical significance of the reported results.’’_
> > > > >
> > > > > The reviewer has repeated this claim multiple times without providing concrete justification. We request that the reviewer act responsibly, like the other reviewers whose questions are concrete and constructive.
> > > > >
> > > > >
> > > > > >_ “I share this concern with one of the reviewers.”_
> > > > >
> > > > > No, the reviewer does not! This review is of far lower quality than that reviewer's, and the concerns raised in this review are not related to theirs. The other reviewer mentioned we might try other statistical analysis to better capture data relationships, which we added in our rebuttal, showing it reinforces our claims. That is clearly an example of constructive and concrete feedback.
> > > > >
> > > > > However, this reviewer's concerns focus only on external peer-reviewed literature and fail to provide any concrete feedback on our paper itself. The two reviews are entirely different.
> > > > >
> > > > > >_ “If area chaird find my score too opinionated they are free to ask for additional emergency review. I am also happy to hear the arguments from other reviewers. Based on the details the authors provide I update my score.”_
> > > > >
> > > > > This review is indeed heavily opinionated in an unfair way.
> > > > > 1. The reviewer has changed their scores completely: from 1-1-1 to 2-2-3. This substantial shift suggests the initial scores were not based on a thorough assessment of our work and were strongly and unfairly opinionated.
> > > > > 2. The reviewer raised the overall score only from 0 to 2, which strongly disagrees with the sub-scores. **This is a clear mismatch, and we request a fair overall score.**
> > > > > 3. The reviewer’s confidence was originally reported as 5. All those major changes and inconsistent behaviors do not justify **a confidence score of 5.** The reviewer’s self-reported confidence does not agree with the subsequent changes in behavior, unfortunately.

---

> > > > > > ### Author Response · Authors · 2025-11-27
> > > > > >
> > > > > > In summary, we believe the reviewer’s comments contain substantial misunderstandings, unsupported judgments, and inconsistencies that do not align with ICLR reviewing standards. We have provided clear evidence and clarifications addressing all points raised. We remain confident in the value, correctness, and significance of our contribution, and we ask for a fair and guideline-consistent evaluation. **We respectfully request that the reviewer (1) provide an updated overall score that accurately reflects the sub-scores, (2) adjust the confidence level to match the substantial score revisions made, (3) stop making negative decisions based on subjective views of the entire field, and (4) engage substantively with our work.**

---

> > > > > > > ### Comment · Reviewer_3PUV · 2025-11-27
> > > > > > > **On the consistency issue**
> > > > > > >
> > > > > > > First, if you find my scores inconsistent, I am happy to change them back to the original ones. I am genuinely interested in the topic and am trying to be constructive, but there seems to be a serious misunderstanding here.
> > > > > > >
> > > > > > > Second, there is no reason for me to criticise the external field. I am just voicing my objection against the attempt to import the standards of rigour accepted in an external field to the field where we work.
> > > > > > >
> > > > > > > Third, you state: "We did not transfer human psychological experiments directly to LLM studies without modification. "
> > > > > > >
> > > > > > > Yet for risk-taking assessment you adapt Columbia Card Task. Appendix H shows a very straightforward application of the CCT to LLM setting. Figure 4 report effects in the opposite direction from "human expectations". Naturally, to have some validity to this Figure one has to refer to some "human expectations" that are statistically validated. Yet (Buelow and Cayton, 2020) report:
> > > > > > > "Two samples of undergraduate student participants (n = 318 Study 1; n = 531 Study 2) completed the IGT, BART, CCT, GDT, and assessment of the big five in a random order. Results of the two studies indicated minimal relationships between performance on the decision making tasks and neuroticism, extraversion, openness, and conscientiousness, but some evidence of a relationship between agreeableness and the CCT in Study 2 only."
> > > > > > > https://www.sciencedirect.com/science/article/pii/S0191886920301203
> > > > > > >
> > > > > > > So, there is mixed evidence on any connection between CCT and Big5 results in humans. So, one can not determine some alignment or misalignment with human expectations in Figure 4. If there is a study with considerable sample size (order of magnitude bigger that the other studies you cite in your work) that shows that there is no evidence for statistically significant association between the scores on big five scales and CCT, your Figure 4 simply misinforms the reader. There are no "human expectations" of statistical significance here. The question of such association is not resolved in psychology.
> > > > > > >
> > > > > > > Or (Hilbig et al. 2013) re-analyse published studies and show that Honesty–Humility, but not Agreeableness predicts active cooperation
> > > > > > > https://psycnet.apa.org/record/2012-34431-001
> > > > > > > see also:
> > > > > > > https://www.sciencedirect.com/science/article/pii/S0092656615000483
> > > > > > > Which raises further questions around Figure 4.
> > > > > > >
> > > > > > > Thus the core statement of the paper is also misinforming the reader:
> > > > > > > "LLMs exhibit clear changes in how they self-report personality traits under different personas, those changes do not consistently manifest in behavior."
> > > > > > >
> > > > > > > I assume you mean "consistency" with respect to the results on humans. Yet, I am trying to show that the results claiming that there is some clear-cut relation between human reports, personality tests and human behaviour are very often inconclusive. Thus there could be no "consistency" or inconsistency definitively observed in your LLM experiments.
> > > > > > >
> > > > > > > You can report that LLMs perform differently on different tasks that assess how they make decision or communicate with users. You can also try to see if the performance on one task correlates (negatively or positively) on the other task. Those tasks might be adapted and based on psychological tasks. Such reports would be interesting for the community and very useful for further research. Yet, mixing in alleged associations between various human scores and human behaviours reduces the value of your work. Instead of staying in the LLM realm and report valid and important results within ICLR community you mix in the results with very low level of statistical validity, produced in a vary different fiel. You also report those results selectively, thus creating certain narrative around your work. Selective reporting of other results from psychology would create a different narrative.
> > > > > > >
> > > > > > > Thus while your experimental work is amazing the main contributions of the paper are erroneous and misinform the reader creating a false impression that there are multiple well reported results on the consistency of psychological tests and human behaviour. The acceptance of such result at ICLR could have significant negative impact on the Machine Learning community, motivating multiple researchers pursue some "alignment" between LLM scores on personality tests and performance on other tasks. Assuming that such alignment is conclusively shown to exist in humans.
> > > > > > >
> > > > > > > I will refrain from engagement with your comments on the quality or substance of my review letting the Chairs to judge it.

---

> > > > > > > > ### Author Response · Authors · 2025-11-27
> > > > > > > > **Response to Reviewer 3PUV's Latest Comments**
> > > > > > > >
> > > > > > > > Dear Reviewer 3PUV,
> > > > > > > >
> > > > > > > > Thank you for your additional comments.
> > > > > > > >
> > > > > > > > In our previous response, we noted two **factual** issues: (1) the scores shifted substantially while the confidence rating remained at 5, and (2) the sub-scores appeared inconsistent with the overall score. We raised these points solely to clarify our understanding of the review. **As the reviewer did not address these factual questions and simply threatened us for reverting the score,** we feel that continued discussion on this matter would not be productive. We are instead bringing this to the attention of the AC/SAC for guidance.
> > > > > > > >
> > > > > > > > Regarding the other content of the review: the concerns of the reviewer are only in the human-trait directions. However, our paper is motivated by a large line of work in the CS community that evaluates LLM “personality” via questionnaires and often assumes that self-reported traits correspond to model behavior. Our contribution is precisely to show that (a) questionnaire-based personality measures are not stable or reliable predictors of actual model behavior, and (b) persona prompting provides limited behavioral control. **Consequently, one or two potentially controversial human trait directions do not affect the validity of our main results or the main message of the paper.**
> > > > > > > >
> > > > > > > > We also note that follow-up work and other reviewers has cited / noted our paper specifically for the instability of questionnaire-based measurements, which aligns with our intended contribution and indicates that this interpretation is consistent with how the research community has understood the work.
> > > > > > > >
> > > > > > > > Given that the reviewer’s score is arbitrary and their comments do not engage with the central contribution of the paper, we believe further back-and-forth will not be helpful, and we defer to the AC/SAC’s assessment.
> > > > > > > >
> > > > > > > > Best :)
> > > > > > > >
> > > > > > > > Submission 26 Authors

---

### Author Response · Authors · 2025-11-27
**Authors' Statement on Review Process Concerns (Reviewer 3PUV)**

**Dear PC, SAC, AC, and public readers,**

Thank you for ensuring a fair review process.

In our response, we noted two **factual** issues with Reviewer 3PUV's evaluation: (1) the scores shifted substantially (1-1-1 to 2-2-3) while the confidence rating remained at 5, and (2) the sub-scores appeared inconsistent with the overall score (2-2-3 vs. overall score 2). We raised these points solely to clarify our understanding of the review. However, instead of addressing them, **Reviewer 3PUV directly threatened to revert the score**, stating: "I am happy to change them back to the original ones."

We are deeply concerned about the reviewer's willingness to arbitrarily change scores with the highest confidence, assign inconsistent sub-scores and overall scores, and invoke the possibility of reverting them in response to our rebuttal pointing out objective factual concerns. This places undue pressure on the authors.

It is indeed disappointing to witness such conduct—including inconsistent scoring and arbitrary threats to revert scores—in the CS and ICLR community. Due to these obvious issues, **we believe continued discussion is not productive, and we hope the AC will intervene appropriately.**

Additionally, the reviewer continues to misunderstand our work. Reviewer 3PUV keeps focusing on the human-trait directions. However, our paper is motivated by a large body of work in the CS community that evaluates LLM "personality" via questionnaires and often assumes that self-reported traits correspond to model behavior. Our contribution is precisely to show that (a) questionnaire-based personality measures are not stable or reliable predictors of actual model behavior, and (b) persona prompting provides limited behavioral control. **Consequently, one or two potentially controversial human trait directions do not affect the validity of our main results or the central message of the paper.**

**Despite stating this repeatedly, the reviewer continues to ignore our clarification and only raises out-of-scope concerns.**

Given that the reviewer's scoring appears arbitrary and their comments do not engage with the central contribution of the paper, we believe further back-and-forth will not be helpful. We defer to the AC/SAC's assessment.

Warm regards,

Submission 26 Authors

---

### Note · Authors · 2026-02-05

**Comment:**

We are withdrawing this submission.

We appreciate the time and effort from the reviewers, AC, SAC, and chairs. After reviewing the final outcome, we have decided to withdraw and pursue publication elsewhere. **We remain confident in the contribution and will incorporate the constructive feedback we received in a revised manuscript.**

We respectfully **disagree** with Reviewer 3PUV’s assessment; for detailed responses to the points raised, we refer readers to our rebuttal and the discussion thread.

We fully understand that this review cycle occurred under unusually challenging and chaotic circumstances, and we appreciate the significant effort required to keep the process running. However, we would still like to note that throughout the entire review and discussion period, the questions and concerns we raised regarding review quality, score consistency, and evaluation criteria did not receive any response from the area chair. Moreover, these concerns were not substantively addressed in the final meta-review.

As a result, we believe these unresolved process issues **materially affected the evaluation**. Given that they remain unaddressed in the final decision, we believe withdrawing is the most appropriate course of action.

Thank you again for your time and consideration.

**Withdrawal Confirmation:**

I have read and agree with the venue's withdrawal policy on behalf of myself and my co-authors.

---

### Meta-Review · Area_Chair_eK2j · 2026-01-06

**Summary:**

1. Reviewer 3PUV

a.1  "Human studies" statements are lacking in quality and statistical soundness

a.2 Most of these papers report some results on very small select groups of people that not even slightly represent the breadth of human experience that LLM is exposed to during the pretraining.

a.3  Many figures have captions that are not legible

a.4 Missing Reference

2. Reviewer 338A

b.1 The statistical methods employed are not well motivated and in some cases the choices seem ill-advised.

b.2 predicting training stage from personality traits seems to confuse the explanans with the explanandum

b.3  statistical modeling of trait stability under repeated prompting

3. Reviewer ajdh

c.1 RQ3 only probes two behaviors . Different methods are not compared

c.2 Limited coverage of imbuement mechanisms

c.3  Extend  analysis to all five behavioral tasks and examine the multi-turn persistence or decay of persona effects

c.4 Activation-level analyses would thus provide a principled, mechanistic check to strengthen the causal link between persona representation and behavioral expression.

**Reviewer Concerns:**

1. Reviewer 3PUV

a.1 and a.2 are not addressed. They are important to the paper.

2. Reviewer 338A

All concerns have been addressed

3. Reviewer ajdh

Only c.1 has been addressed. c.2 and c.4 have not been addressed.

**Reviewer Scores:**

1. Reviewer 3PUV. As the authors do not want to address these critical issues. The reivewer will not change the score.

2. Reviewer 338A will not change the score

3. Reviewer ajdh will not, and may decrease the score.

---

### Decision · Program_Chairs · 2026-01-26

Reject